# Probing the active site in single-atom oxygen reduction catalysts via operando X-ray and electrochemical spectroscopy

Hsiang-Ting Lien [1,2,8], Sun-Tang Chang[3,8], Po-Tuan Chen[1,2,4,8], Deniz P. Wong[5], Yu-Chung Chang[1,3,6], Ying-Rei Lu[6,7], Chung-Li Dong[7], Chen-Hao Wang [3], Kuei-Hsien Chen [1,5✉] & Li-Chyong Chen [1,2✉]

Nonnoble metal catalysts are low-cost alternatives to Pt for the oxygen reduction reactions (ORRs), which have been studied for various applications in electrocatalytic systems. Among them, transition metal complexes, characterized by a redox-active single-metal-atom with biomimetic ligands, such as pyrolyzed cobalt–nitrogen–carbon ($Co–N_x/C$), have attracted considerable attention. Therefore, we reported the ORR mechanism of pyrolyzed Vitamin B12 using operando X-ray absorption spectroscopy coupled with electrochemical impedance spectroscopy, which enables operando monitoring of the oxygen binding site on the metal center. Our results revealed the preferential adsorption of oxygen at the $Co^{2+}$ center, with end-on coordination forming a $Co^{2+}$-oxo species. Furthermore, the charge transfer mechanism between the catalyst and reactant enables further Co–O species formation. These experimental findings, corroborated with first-principle calculations, provide insight into metal active-site geometry and structural evolution during ORR, which could be used for developing material design strategies for high-performance electrocatalysts for fuel cell applications.

[1] Center for Condensed Matter Sciences, National Taiwan University, Taipei, Taiwan. [2] Center of Atomic Initiative for New Materials, National Taiwan University, Taipei, Taiwan. [3] Department of Materials Science and Engineering, National Taiwan University of Science and Technology, Taipei, Taiwan. [4] Department of Vehicle Engineering, National Taipei University of Technology, Taipei, Taiwan. [5] Institute of Atomic and Molecular Sciences, Academia Sinica, Taipei, Taiwan. [6] National Synchrotron Radiation Research Center, Hsinchu, Taiwan. [7] Department of Physics, Tamkang University, Tamsui, Taiwan. [8] These authors contributed equally: Hsiang-Ting Lien, Sun-Tang Chang, Po-Tuan Chen. ✉email: chenkh@pub.iams.sinica.edu.tw; chenlc@ntu.edu.tw

The development of renewable energy has accelerated the transition from fossil fuels to next-generation power sources. Among the numerous renewable energy sources, proton exchange membrane fuel cells (PEMFCs) offer high-power and clean energy that could satisfy energy demands. However, challenges remain in PEMFC development. A major obstacle is the oxygen reduction reaction (ORR) at the cathode, which is a rate-determining step and requires enzyme and catalyst assistance. However, the precious Pt metal is required as a catalyst, hindering the commercial spread of PEMFCs. Tremendous efforts have been made in developing nonprecious metal–nitrogen–carbon (denoted as M–$N_x$/C) complexes to reduce related costs; they are prepared using simple pyrolysis. These substitutes have displayed promising progress as a replacement for Pt[1–8]. Despite considerable improvements to M–$N_x$/C electrocatalysts' activity and stability since they were first synthesized[1], the key steps in catalyzing oxygen molecules into water through electron transfer remain unclear. Catalyst characterizations using ex situ techniques are limited in terms of the understanding of catalytic behavior that they provide[4,6]. Therefore, direct probing of the active site under operando conditions is imperative for constructing a detailed model of the catalyst state at the macroscopic, microscopic, and even atomic levels in action[9].

Direct spectroscopic observations of M–$N_x$/C catalysts during reaction conditions are extremely difficult because of the single-atom size of the active site. However, advancements in in situ X-ray absorption (XAS) techniques have enabled scientists to selectively probe the immediate environment of a few metal atom catalysts[10–12]. The geometric information and the electronic states, such as oxidation state and local symmetry, could be revealed using a metal X-ray absorption near edge structure (XANES) and extended X-ray absorption fine structure (EXAFS)[13]. In situ and operando studies have occasionally employed hard XAS (h-XAS) because of the relatively simple experimental setup that functions in an ambient environment. XAS analysis is rarely performed using soft XAS (s-XAS) ranges because of the difficulties of measuring in a vacuum environment and probing under liquid phase conditions. These experimental challenges can be overcome using photon-in–photon-out techniques in the s-XAS range with an information depth of a few hundred nanometers. During related catalytic reactions, the d-block transition metals are crucial in catalysis because of the unfilled d-orbitals. Metal L-edge probing of the 2p electrons to unfilled d-orbitals is more sensitive to the d electrons than the $1s \rightarrow nd$ transitions in the pre-edge condition of the metal K-edge[14,15]. Therefore, performing measurements in the s-XAS range could be used to understand catalytic behavior. Although XAS has been successfully used to study changes during redox

reactions, these studies and mechanistic interpretations related to the M–$N_x$/C catalysts remain scarce. Density functional theory (DFT) studies on vitamin B12 materials have revealed the relationship between geometries and electronic structures of cobalt (Co)–corrin centers[4]. However, the functions of these materials during ORR is unclear. Performing and analyzing the operando spectra of the catalytic Co center based on theoretical calculations could provide a deeper understanding of the ORR procedure.

Here we show, a spectroelectrochemical cell with a biomimetic electrocatalyst electrode, pyrolyzed vitamin B12, is developed for operando XAS measurements in a wide X-ray range. We provide a comprehensive report on the reactivity and catalytic sites of $O_2$ reduction on an active Co site through operando XAS using electrochemical impedance spectra (EIS) to obtain electronic, geometric, and structural information under catalytic conditions. This technique enables the investigation of changes in the Co metal site from the mesoscopic to the atomic level and the evolution of the $Co^{2+}$-oxo intermediate states under different bias voltages. Corroboration of the experimental observation with theoretical models may elucidate the chemical reaction mechanisms[16,17]. Combining the EXAFS, $\Delta\mu$ technique, and EIS data, possible adsorbates on the electrocatalyst surface were assessed, providing a method to investigate the intermediate state during the catalytic reaction. Furthermore, the hybridization of a Co d-orbital (also coupling with C) with an O p-orbital formed an antibonding state with oxygen, which effectively transferred charge from the low valent $Co^{2+}$ to oxygen $\sigma^*$, thereby forming $Co^{2+}$-oxo species. These processes reduce the energy barrier for electrochemical $O_2$ separation. The measurement techniques proposed in this study will be applicable in clarifying other complex multielectron catalytic processes, such as $CO_2$ reduction and hydrogen evolution reaction.

## Results and discussion

**Electrochemical analysis.** Linear sweep voltammetry (LSV) curves were measured for the electrocatalyst, biased under different potentials in an $O_2$-saturated 0.1-M $HClO_4$ solution, to evaluate the reduction efficiency in the potential range of 1.2–0 V (compared with reversible hydrogen electrode (RHE)), as presented in Fig. 1a, during which the ORR process exhibited three conditions, the mixed diffusion-controlled condition (at ~1.1–0.9 V), kinetic-dominant condition (at 0.8–0.4 V), and mass transport condition (at 0.3–0 V). Nyquist plots were recorded under constant potential in a frequency range of 1 Hz–1 MHz, as presented in Fig. 1b. EIS provides information regarding the kinetic and mass transport properties of the catalyst, which could reveal the relationship between the electrical measurements and the chemical changes that occurred in the reaction processes. The

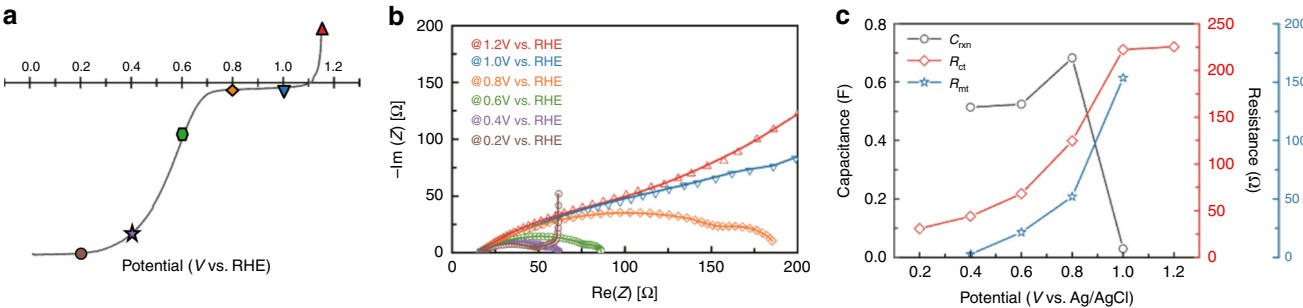

**Fig. 1 In situ electrochemical spectrum by home-built spectroelectrochemical cell. a** The LSV curve of the py-B12 **b** Nyquist plots of ORR on py-B12 with concurrent XAS measurement. **c** The EIS fitting spectra of ORR recorded with concurrent XAS measurement. All the signals were collected on a home-built spectroelectrochemical cell with a catalyst loading of 100 mg cm$^{-2}$ and under $O_2$-saturated 0.1-M $HClO_4$ solution.

high-frequency intersections at the real axes the Nyquist plots represent the contact resistance of the catalyst on the electrode. The resistor–capacitor circuit displayed an arc or semicircular trace, which reflected the double-layer capacitance of the catalyst layer ($C_{DL}$), whereas the interfacial mass and charge transfer of oxygen on the catalyst's active sites were reflected in the charge transfer resistance ($R_{ct}$), mass transfer resistance ($R_{mt}$), and capacitance of the reaction process ($C_{rxn}$). The $C_{DL}$ was independent from applied potential, suggesting that it originated from the capacitor component of the catalyst layer. Furthermore, applied potential dependence was observed for $C_{rxn}$, which was related to the adsorption of oxygenate-based intermediates. Diffusion-dominated processes were observed at high (>1.2 V) and low (<0.2 V) voltages. At 1.2 V, gaseous oxygen diffused through the electrolyte liquid film to the catalyst sites. The mass transport condition at 0.2 V is attributable to the slow diffusion of oxygen through the backing layer, the back diffusion of water, or the diffusion of water in the catalyst layer[18–20].

A simple fitting model from the proposed equivalent circuit was used to extract the various charges and mass transfer impedances of the catalyst layers of the in situ cell. The fitted EIS data with the respective equivalent circuits under different bias conditions are displayed in Supplementary Table 1. Figure 1c displays the $R_{ct}$, $R_{mt}$, and $C_{rxn}$ plotted as a function of applied potential. Under the mixed diffusion-controlled condition (at ~1.1–0.9 V), the oxygenated reactants diffused to the active sites. Therefore, the resistances of the charge transfers were higher. The

reduction of the $R_{ct}$ and $R_{mt}$ under the kinetic-dominant condition (at 0.8–0.4 V), under which the catalyst was likely covered with oxygenated adsorbates, was rapid compared with the reduction at 1.0 V. The $R_{ct}$ and $R_{mt}$ reached a stable value at 0.4 V. Studies of Pt/C during ORRs have suggested that the adsorption of various oxygenate species at the electrode surface enables fast transport of electrons during ORR, resulting in a considerable reduction of $R_{ct}$ and $R_{mt}$.

**Operando Co K-edge XANES/EXAFS under bias**. The LSV revealed that the ORR is divided into three conditions. Operando Co K-edge XANES/EXAFS studies were performed to further examine the electronic structural changes of the py-B12 electrocatalyst under precatalytic to catalytic conditions. Figure 2a illustrates the normalized intensity of the Co K-edge at various bias voltages. Co K-edge spectra mainly describe the transitions of 1s to 4p, typically attributed to core-level spectroscopy, whereas the dipole-forbidden 1s to 3d transitions are less intense in the K-edge spectra. However, the pre-edge transition peak (1s to 3d) of the Co K-edge can be attributed to high plane $D_{4h}/S_4$ symmetry, which indicates the formation of a square planar Co–$N_x$ structure[21]. The oxidation states of the Co atoms in py-B12 can be determined from the $E_0$ positions at the pre-edge of the Co K-edge XANES, which were obtained by calculating the highest point and the inflection point of the first- and second-derivative spectra, respectively.

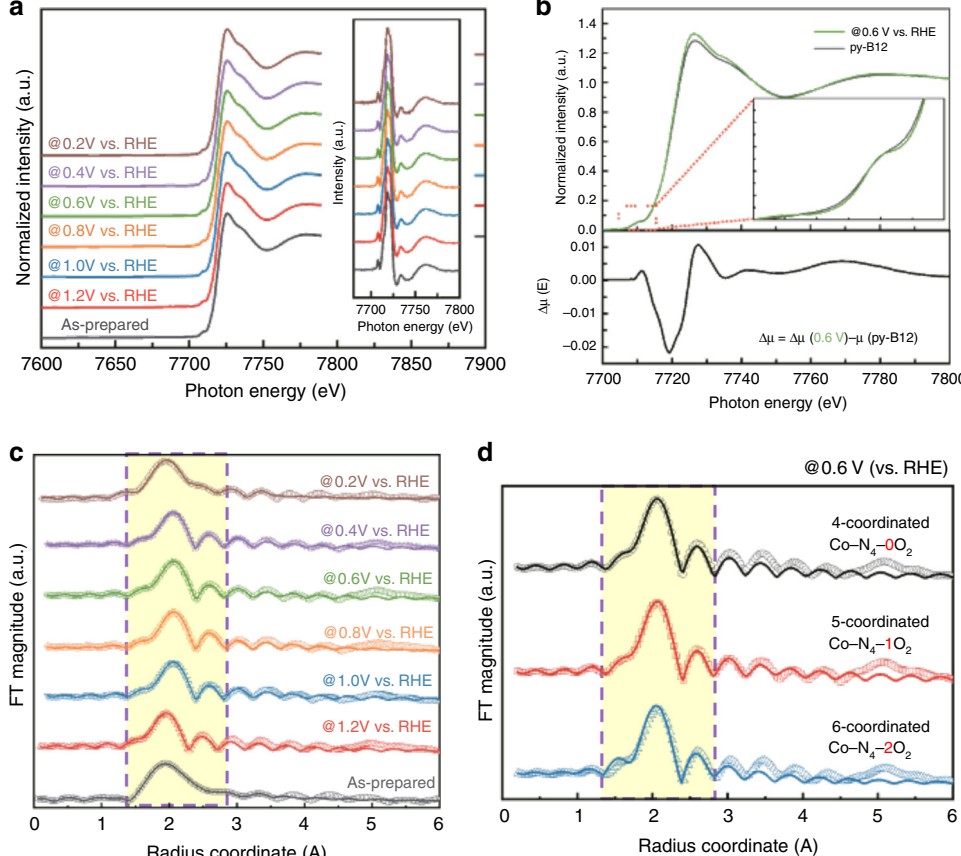

**Fig. 2 In situ Co K-edge X-ray absorption. a** Co K-edge XANES spectra of py-B12 at various operando biases (insert shows the first-derivative spectrum). **b** The XANES and Δμ-XANES spectra obtained the sample at a bias of 0.6 V (vs. RHE) and in the as-prepared state. **c** The phase-corrected Fourier transforms EXAFS ($k^3$-weighted) obtained at various operando biases. **d** The EXAFS spectrum, which was obtained under a bias of 0.6 V (vs. RHE), is fitted with three different models. The region highlighted in soybean color was the fitting region. All the signals were collected on a home-built spectroelectrochemical cell with catalyst loading of 100 mg cm$^{-2}$ and under O$_2$-saturated 0.1-M HClO$_4$ solution.

**Table 1 $E_O$ edge from XANES and EXAFS fitting parameters for sample under different conditions: as-prepared, under precatalytic (1.2 V), and catalytic state (1.0–0.4 V).**

| Sample | $E_O$ of Co K-edge (eV) | Coordinated number CN | EXAFS R (Å) | $\sigma^2$ |
|---|---|---|---|---|
| As-prepared | 7718.6 | Co-N: 3.99 | 1.95 | 0.0015 |
| 1.2 V vs. RHE | 7718.5 | Co-N: 3.99 | 1.96 | 0.0016 |
| 1.0 V vs. RHE | 7718.5 | Co-N: 4.02 | 2.01 | 0.002 |
| | | Co-O: 1.00 | 2.35 | |
| 0.8 V vs. RHE | 7718.5 | Co-N: 4.01 | 2.01 | 0.0036 |
| | | Co-O: 0.99 | 2.35 | |
| 0.6 V vs. RHE | 7718.4 | Co-N: 3.98 | 2.02 | 0.0025 |
| | | Co-O: 1.01 | 2.36 | |
| 0.4 V vs. RHE | 7718.6 | Co-N: 4.01 | 2.02 | 0.0028 |
| | | Co-O: 1.01 | 2.36 | |
| 0.2 V vs. RHE | 7718.5 | Co-N: 3.99 | 1.95 | 0.0048 |

**Table 2 EXAFS fitting parameters for sample under catalytic state with O-based adsorbates: amount of O varied from 0 to 2.**

| Fitting condition (@0.6 V vs. RHE) | Coordinated number CN | EXAFS R (Å) | $\sigma^2$ |
|---|---|---|---|
| 4-coordinated ($0O-CoN_4$) | Co-N: 35.6 | 1.81 | 0.12 |
| 5-coordinated ($1O-CoN_4$) | Co-N: 3.98 | 2.02 | 0.0025 |
| | Co-O: 1.01 | 2.36 | |
| 6-coordinated ($2O-CoN_4$) | Co-N: 1.20 | 1.87 | 0.33 |
| | Co-O: 7.61 | 2.12 | |

The rising-edge XANES peaks at ~7718.5 eV were assigned to the $Co^{2+}$ state, which appeared to be the predominant oxidation state under various potentials in ORR[4].

The $\Delta\mu$-XANES analysis technique is a surface-sensitive technique used to assess surface chemistry involved in catalysis, such as weakly-bound adsorbate interactions[22]. Figure 2b presents the Co K-edge XANES data of the py-B12 electrocatalyst under two conditions, which were as-prepared and operated at 0.6 V (compared with RHE). The pristine sample was adsorbate-free with a square planar $Co-N_x$ configuration, whereas the metal center was bonded with oxygen with a $Co-N_x-O_{ads}$ structure for the sample operated at 0.6 V. The $\Delta\mu$-XANES spectra were subtracted according to the equation

$$\Delta\mu = \mu(Co - N_x - O_{ads}) - \mu(Co - N_x). \quad (1)$$

Therefore, the positive peak in the pre-edge condition reflected the change in Co ($1s$ to $3d$). A forbidden transition was observed at ~7710 eV because the catalytic reaction occurred at the unfilled $3d$ orbital[23,24]. The positive peak feature could only be observed when the adsorbed oxygen atom was placed in the edge-on position of the metal center. Moreover, the negative peak at ~7720 eV indicated a charge transfer from neighboring Co atoms to the adsorbed oxygen. The Co K-edge XANES data indicated that the oxidation state of Co only changed partially.

The Co K-edge EXAFS spectra were recorded ex situ on the py-B12 catalysts (as-prepared), and operando XAS spectra were recorded in the precatalytic and catalytic states (under bias). The EXAFS fitting parameters are displayed in Fig. 2c and Table 1. For the as-prepared py-B12, the Co–N coordinated bonds at 1.95 Å had a coordination number (CN) value of 4. Under 1.2-V (vs. RHE) precatalytic conditions, the EXAFS data with the best fit were those for the Co–N bond at 1.96 Å, with a CN of almost 4. The bond was slightly elongated after a certain potential was applied (1.0–0.4 V vs. RHE), and the best fit was obtained at 2.01 Å for the Co–N coordinated bond, with a CN of approximately 4. This result suggests a Co–N bond elongation of ~3% compared with the precatalytic state. Presumably, the elongation of the Co–N bond was caused by the distortion of the original square planar $Co-N_4/C$ configuration, which arose from the displacement of the Co atom from the $Co-N_4/C$ plane induced by oxygen-based adsorbates, such as $O_2$, O–O, OH, or $H_2O$. An additional Co–O coordinated bond with a CN of 1.0 was observed after a certain potential was applied (1.0–0.4 V vs. RHE), which indicates that Co bonded with one oxygenated species at a time. The DFT calculations suggested a fitting parameter of 2.35 Å as the physical adsorption of the $O_2$ molecule on Co. Moreover, a shoulder at 1.84 Å was assigned to the length of Co bonded with a separated O atom. Notably, this bond length was shorter than the typical Co–O bond of bulk CoO (1.89 Å), as illustrated by a reference bond in Supplementary Fig. 1.

Moreover, the DFT calculations indicated the presence of a chemisorption bond length of Co and dissociative OO adsorption located at 1.96 Å, which is behind the broad peak of 2.01 Å.

Further insight into the local environment during the reaction can be obtained by fitting the EXAFS data with different models to identify the O-based species adsorbed on the py-B12 plane using $Co-N_4$ with different O amounts from 0 to 2, as illustrated in Fig. 2d. In the r-space, the data were fitted between 1.2 and 2.8 Å. The fitting data are reported in Table 2. For the $0O-Co-N_4$ and $2O-Co-N_4$ models, the fitting CNs were 35.6 and 1.20 with improbable Co–N shell distances of 1.81 and 1.87 Å, respectively. By contrast, the corresponding fitting data for $1O-Co-N_4$ were a CN of 3.98 and Co–N of 2.02 Å, which were the most plausible structural parameters. Studies have indicated that the $Co^{2+}$-dioxygen complex also forms 1:1 adducts with low $O_2$ affinity[25,26]. Moreover, a side-on geometry was omitted for the binding of $O_2$ to a $Co^{2+}$ or $Fe^{2+}$ porphyrin because this would likely lead to an $M^{2+}$ side-on superoxo species[27,28]. The Co–N bond distortion originated from the Co atom being displaced from the $Co-N_4/C$ plane because of the oxygen-based adsorbate. In summary, the CN was 5 at different applied potential, including four bonds to N and one bond to O. Therefore, a single oxygen-based intermediate is adsorbed on the reactive Co atom surrounded with 4N atoms during evolution under applied potentials of 1.0–0.4 V. Although the environment surrounding the central Co atom can be deduced from EXAFS, the CN is merely the first shell information. The information regarding multiple shells is still unclear. The EXAFS do not reveal which oxygen species are bound to the Co atom. Therefore, the different oxygen species bonded to the central $Co-N_4/C$ were investigated using theoretical calculations.

**Operando Co $L_{3,2}$-edge NEXAFS of py-B12 electrocatalysts.** Co $L_{3,2}$-edge NEXAFS studies were conducted under operando conditions to analyze the ORR of the py-B12 electrocatalyst at the catalyst–electrolyte interface from the bulk scale to the molecular and even atomic levels. In general, transition $d$-block metals are crucial in catalysis because of unfilled $d$-orbitals. Co $L_{3,2}$-edge NEXAFS describing the transitions from $2p$ to $3d$ unfilled orbitals can be used to monitor the oxidation state and charge transfer between the catalyst and reactant. Figure 3a demonstrates that Co $L_{3,2}$-edge NEXAFS changed under various potentials in the ORR. The characteristic peaks of the $Co^{2+}$ observed in both the pristine sample and the corresponding sample at 1.2 V (vs. RHE) indicated that the corresponding sample was under the precatalytic condition in which oxygen reactants diffused to the active sites without chemical reaction. Under an applied bias of 1.0–0.4 V, the Co peak displays a progressive upshift to a higher photon energy (<1 eV), which indicated that the chemical reactions occurred in this bias condition. The reference data for the $Co^{2+}$

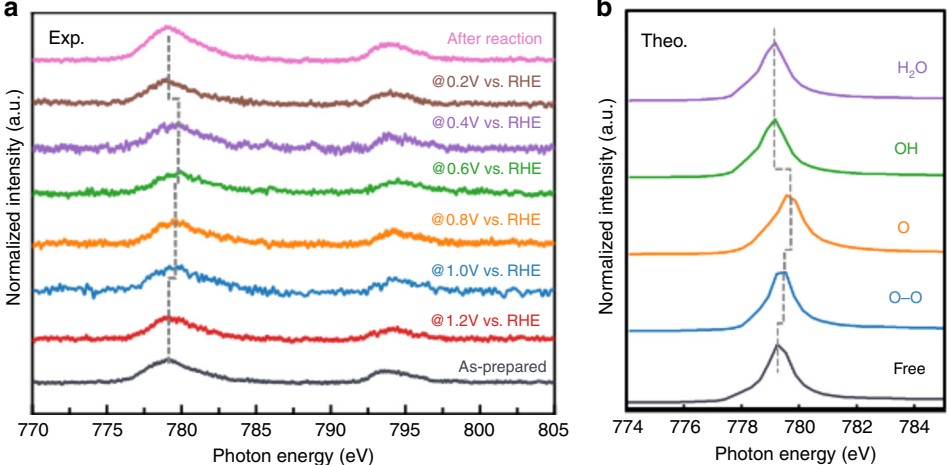

**Fig. 3 In situ experimental and theoretical calculation data of Co L-edge X-ray absorption. a** Co $L_{3,2}$-edge X-ray absorption near edge spectra (XANES) of the py-B12/C catalyst at various operando biases. **b** FEFF calculated Co L-edge with different oxygenate species.

and $Co^{3+}$ L-edge (as illustrated in Supplementary Fig. 2) suggest that Co was moderately oxidized from 2+ to $2 + \delta$ $(0 < \delta < 1)$ during the catalytic reaction. The peak shifted from the +2 valence number region to a higher valence number region, suggesting that the catalytic process was accompanied by a partial electron transfer from the 3d orbital of the Co active site to adsorbed oxygen species. At 0.2 V, the peak returned to its original position, which indicates that the final product was desorbed from the Co site.

Figure 3a reveals the experimental XANES. We observe that the peak shifted at a different applied potential. The $L_{3,2}$-edge NEXAFS studies revealed that Co was partially oxidized under catalytic conditions in this class of material. The shift can be attributed to the summation of the different $Co^{2+}$-oxo intermediates. First-principle calculations were then performed to obtain further insight into the $Co^{2+}$-oxto geometries and electronic structures. Moreover, the analysis of operando XANES spectra revealed the evolution of the $Co^{2+}$-oxo intermediate state (Fig. 3b). The findings also accorded with the EIS and EXAFS data.

$O_2$ molecules can undergo physical adsorption on a py-B12 of neutral charge with one O atom at the active Co site with a length of 2.20 Å. The higher charge and mass transfer resistance at the precatalysis condition demonstrated that the onset of $O_2$ adsorption was diffusion-controlled. After the bias was applied, the dissociative adsorption of OO on a negatively charged py-B12 occurred spontaneously[4]. The dissociative OO was chemically adsorbed on the Co site with an O–Co bond of 1.96 Å. Furthermore, the dissociative OO adsorption facilitated O–O separation over a mild barrier. In the separation of O–O, one O bonded with Co and the other O bonded with a C atom of py-B12 with an O–Co bond length of 1.84 Å. The FEFF calculations suggested that OO and O bonding with Co induced changes in the Co oxidation state and resulted in the upshifted Co L-edge, as illustrated in Fig. 3b. The FEFF results suggested that the adsorption of an O atom on Co led to the largest shift. The ORR led to a 1.0-eV peak upshift of the Co L-edge under a bias voltage of 0.8–0.4 V. The upshift was attributable to a charge transfer from cobalt to oxygen because of the evolution of the $Co^{2+}$-oxo intermediate state, which resulted in a lower transfer resistance, as demonstrated by the EIS data. The reaction of the separated product with the H atoms on the cathode can readily produce two adsorbed hydroxyl radicals, HO–Co, which can further produce two $H_2O$ molecules. Under the 0.3–0 V potential condition, OH and $H_2O$ occupied the active Co site. The OH and $H_2O$ products did not change the oxidation state of Co, which was reflected in

the lack of shift of the Co L-edge. However, further mass transfers would be required before the active Co sites could be used again. Because of the small interaction between $H_2O$ and the Co site, $H_2O$ may be desorbed from the catalyst surface.

The sequence of $O_2$, O–O, OH, or $H_2O$ adsorption represents the four-electron ORR process. The operando Co L-edge NEXAFS provided details on the reactions following the four-electron procedure. The two-electron ORR case was also considered. The formation of the $Co^{2+}$-oxo intermediate states, such as $O_2$, $H_2O_2$, OH, or $H_4O_2$ adsorption, did not induce changes in the oxidation state in the theoretical calculation. The details of the FEFF calculations are reported in Supplementary Fig. 3.

**Electronic structure variation in the catalytic process**. In the four-electron ORR process, the direct bond breaking of O–O is the major rate-determining step. However, the dissociation of OO on the singe negatively charged py-B12 can occur spontaneously over a mild barrier producing O–Co and O–C. The mechanism can be interpreted by frontier orbital theory based on our operando findings.

The model in Fig. 4 displays the hybridizations of local molecular orbitals (MOs) of the active Co center and $O_2$ reactant. The highest occupied MOs of oxygen (left side) and square planar Co–$N_4$/C (right side) are illustrated in the top-right panel. According to the formation of the expected square planar Co–$N_4$ structure, d electron crystal field splitting involves the $d_{x^2-y^2}$ orbital, which has the highest energy state, followed by the $d_{xy}$ orbital, $d_{z^2}$ obrbital, and finally $d_{xz}$ and $d_{yz}$ orbitals. The energy splitting difference of $d_{x^2-y^2}$ and $d_{xy}$ is always large, even in a weak field[29,30]. Therefore, square planar complexes typically have low spin. An electron of $Co^{2+}$ $d^7$ occupied the $d_{xy}$ orbital, which might transfer into the antibonding state.

The hybridized orbitals are described in the bottom-left scheme of Fig. 4. During ORR, the electrons of $O_2$ coupled with the d-orbital of Co. One low spin unfilled a $3d_{xy}$ electron of Co bonded with an antibonding orbital of oxygen, forming an antibonding orbital of $\sigma^*(d_{xy} + \sigma_{2p})$. Therefore, the three highest occupied MOs were in the antibonding state that weakens the bonding of oxygen. The upshift of the Co $L_{3,2}$-edge further demonstrated the orbital hybridization between Co–$N_4$/C and oxygen. The resultant energy gap between these two orbitals and the other three metal d-orbitals may be sufficient to overcome the energy involved in spin pairing and thus lead to five-coordinate low-spin species[31]. Moreover, this orbital hybridization of Co–$N_4$/C with oxygen engendered a charge transfer to form $Co^{2+\delta}$ in the

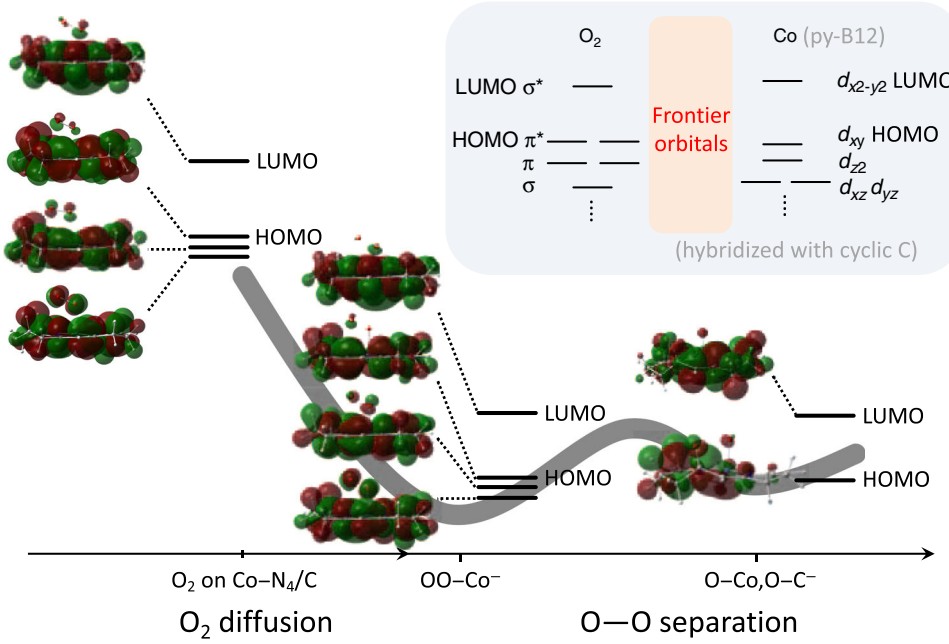

**Fig. 4 Molecular orbital description of Co–N$_4$/O$_2$ interaction.** Simulated molecular orbital in accordance with the four-electron ORR process. The $d^7$ electron crystal field orbital and a hybridization orbital, along with the oxygen electron ($\sigma^*$ and $\pi^*$) and a schematic model of the Co–N$_4$/C–O$_2$ bond configuration. The potential curve is referred to ref. [4].

oxygen $\sigma^*$, and $\pi^*$ orbital of O$_2$ to form an OO$^{\delta+}$ species. According to simulations, O$_2$ was adsorbed on the Co–py-B12 surface as a negatively charged O$_2^-$ intermediate, which was expected to exist in abundance in the cathode under ORR conditions[4]. Therefore, the Co$^{2+}$ center could be more easily oxidized because of the higher O$_2$ affinity[25,26]. The higher electron density was likely to promote the hydrolysis of the Co–O$_2$ species, resulting in the formation of 4e$^-$/4H$^+$ H$_2$O rather than 2e$^-$/2H$^+$ H$_2$O$_2$.

The analysis of the L-edge data on the pristine B12 and as-prepared py-B12 (as illustrated in Supplementary Fig. 4 and Supplementary Table 2) revealed that the central metal oxidation states were Co$^{3+}$ and Co$^{2+}$, respectively. The operando study of the py-B12 under an applied potential of 0.6 V demonstrated an orbital hybridization of Co–N$_4$/C with oxygen, as revealed by a multipeak fit to the Co L-edge spectrum (Supplementary Fig. 4). These findings demonstrate that the L$_3$-edge is useful for experimentally investigating the electronic structural description of the Co–O$_2$ active center; similarly, we can apply this technique to other Co–O$_2$ or Fe–O$_2$ complexes[32,33]. These experiments are highly useful for improving our understanding of factors that determine oxygen affinity for cobalt systems and, by extrapolation, for other transition metal-based stems.

In summary, the combination of in situ XAS and EIS enabled the investigation of the electronic and structural changes that occur in biomimetic nonprecious metal macrocyclic electrocatalysts under actual working conditions. This approach also brings us closer to understanding the catalyst's active site, especially regarding oxygen interaction with the single-metal-atom with biomimetic ligands. From the combination of the potential sweep during XAS studies (Co K-edge), we observed that the formation of a Co–O bond extended to the Co–N bond, suggesting a charge transfer effect between the reactant and the catalyst. Furthermore, insight into the splitting in the $d_{xy}$ orbital of the metal and its interaction with the $\sigma^*$ and $\pi^*$ orbital of O$_2$ can be revealed by analyzing the Co L-edge. The experimental findings were supported by simulations of O$_2$ adsorbed on the Co–py-B12 surface as a negatively charged O$_2^-$ intermediate, resulting

in the formation of a four-electron process, 4e$^-$/4H$^+$ H$_2$O, rather than 2e$^-$/2H$^+$ H$_2$O$_2$. These new findings regarding the electronic structure of the metal-organic complex with catalysis offered opportunities for greater fundamental understanding, mechanistic insights, and possible design strategies for technologically relevant reactions in the future.

## Methods

**Materials and electrocatalyst ink preparation.** Cyanocobalamin (vitamin B12, VB-12) and perchloric acid obtained from Sigma-Aldrich (purity > 98%) were used as the electrocatalyst and electrolyte, respectively. Carbon black (Vulcan XC-72R) was used as the electrocatalyst support. The catalyst ink preparation was described in our previous work[4–7]. In brief, 0.2 g of VB-12 was mixed with 0.3 g of XC-72R in water with vigorous stirring and ultrasonic agitation for 30 min, which formed a uniform slurry solution. The powder was then dried in a rotary evaporator. The dried mixture was placed in an aluminum oxide boat, introduced into a quartz tube furnace, and pyrolyzed at 700 °C at a ramping rate of 20 °C min$^{-1}$ surrounded by N$_2$ for 2 h. The product is denoted as py-B12 with an active complex of Co–N$_x$/C. The electrocatalyst ink was prepared by mixing 160 mg of the catalyst with 20 mL of deionized water. The slurry ink was then agitated in an ultrasonic water bath for 30 min. A drop cast of 20 μL of the ink and 5 μL of 0.1 wt% Nafion solution was used to coat the electrode and air-dried at room temperature before further electrochemical and spectroscopic studies. A brief review of the ex situ characterizations is presented in Supplementary Fig. 5.

**Operando X-ray absorption and electrochemical impedance measurements.** The XAS measurements were performed at the National Synchrotron Radiation Research Center (NSRRC, Hsinchu, Taiwan). The operando X-ray spectroscopic studies, s-XAS and h-XAS, were conducted at 60–1250 and 4.8–14.2 keV, respectively. The s-XAS measurements for the Co L$_{3,2}$-edge (770–810 eV) and h-XAS measurements for the Co K-edge (7.709 keV) were performed using the NSRRC beamlines BL20 A1 and BL17 C1, respectively. The measurements were performed using a homemade three-electrode spectroelectrochemical cell. Pt was used as the counter electrode and Ag/AgCl was used as the reference electrode. Electrochemical measurements were connected to a potentiostat–galvanostat instrument (Bio-Logic SP-240) and measured in a 0.1-M O$_2$-saturated HClO$_4$ electrolyte solution. The electrocatalyst was drop-coated on the desired working electrode. The experimental setup is illustrated in Supplementary Fig. 6. Different working electrodes were used because of the different measuring environments for s-XAS and h-XAS. For h-XAS, an Au mesh of ~1 cm$^2$ was used, and the signal was collected in transmission mode. Pure metal foil was used for energy calibration. For h-XAS, XANES- and EXAFS-related ranges were collected to reveal the changes in electronic structure, coordination environment, and oxidation state during ORR. For s-XAS (NEXAFS), a working electrode composed of 100-nm-thick Si$_3$N$_4$ with a

0.1-mm$^2$ window on a 500-μm Si frame was used to separate the ambient and vacuum environments during measurement. The window was coated with 2 nm of Cr and 10 nm of Au to provide metallic contact, following which the electrocatalyst was drop-casted. NEXAFS measurements were calibrated using a reference metal oxide powder and fluorescence yield data were collected. The presented data were merged from 10 spectra using the on-the-fly scanning system. Each spectrum was collected for 40 s[34]. For instance, pristine B12, the Co L-edge is obtained at a depth of 1 μm, as well as the K-edge is obtained at ca. 100 μm. Therefore, the spectral signal is not only the surface adsorption state, but also part of the bulk information. K-edge contains a large proportion of bulk information, which is not conducive to the analysis of adsorbates. Therefore, the $\Delta\mu$ analytical technique of K-edge is used to discuss the adsorption of py-B12.

Before any operando measurement commenced, an ORR activity test was performed using LSV at 0–1.2 V for comparison with the RHE at a scan rate of 10 mV s$^{-1}$ in the same spectroelectrochemical cell. Operando ORR measurements were performed at a potential range of 0–1.2 V for comparison with Ag/AgCl. The potentiometric experiment was performed to collect XAS data under different potentials. Spectroscopic techniques limited the electrochemical experiments through application of a constant potential. Furthermore, EIS data were collected during X-ray-based spectra collection. The EIS spectra were recorded under an AC amplitude of 10 mV at a frequency of 1 MHz–1 Hz under various electrochemical potentials. LSV curves and EIS spectra were obtained using a potentiostat and impedance analyzer (Bio-Logic SP-240). The EIS spectra were fitted to the equivalent circuit using commercial software (EC-Lab, Bio-Logic Science Instruments).

**XAS data reduction and analysis**. The XANES/EXAFS and NEXAFS spectra were analyzed using Demeter[35]. Both the pre-edge and post-edge backgrounds were subtracted from the XANES spectra for normalization in Athena and curve fitting in Artemis. For the Co K-edge XANES spectra, data were referenced to the Co foil data, for which an $E_0$ value of 7709 eV was used. Normalization was performed at 7560–7690 and 8360–8560 eV. For the Co L$_{3,2}$-edge NEXAFS spectrum, normalization was performed at 770–775 and 800–805 eV.

**Theoretical approach**. DFT calculations combined with multiple scattering simulations[11,36] were used to determine the electronic structure of M–N$_x$/C during the X-ray-based spectra collection, measured by XANES under various bias voltages. The central py-B12 structure was modeled by removing all branched chains of VB-12 and then terminating with an H atom[4]. Structural optimizations of the py-B12 model with oxygen-based adsorbates, such as O$_2$, O–O, OH, or H$_2$O, were performed using DFT at the B3LYP level[37,38] with the LANL2DZ basis set[39,40]. The calculations had one negative charge to simulate the current at the cathode, except the O$_2$ molecule, which was only physically adsorbed on py-B12 with a neutral charge. In principle, the single negatively charged model can be formed by the adsorption of O$_2^-$ on py-B12, capturing an electron by the neutral intermediate OO–Co, or the adsorption of a neutral O$_2$ on negatively charged py-B12, which is thought to exist in abundance in the cathode under ORR conditions. All the stationary points were identified for local minima using vibrational analysis at the same level. The electronic structure calculations were performed using Gaussian 16[41].

The features in the Co L-edge spectra and how they linked to changes in the electronic structure and evolution of Co$^{2+}$-oxo intermediate states during ORR were then identified. The results of the measurements of py-B12 under bias were compared with the calculation results. XANES calculations were then performed using DFT-optimized structures to theoretically examine Co L-edge XANES spectra related to oxygen-based adsorbates on the Co–corrin cluster. XANES calculations of Co were performed using the FEFF8 code[42]. The self-consistent potential and full multiple scattering were calculated at a 5.0-Å radius. To compare the experimental and calculated spectra, a rigid shift to a higher energy of 2 eV was applied to each calculated spectrum.

## Data availability

The data that support the findings of this study are available from the corresponding author upon reasonable request.

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

## Acknowledgements

The authors thank the staff at BL 17C and BL 20A of National Synchrotron Radiation Research Center (NSRRC) for helping in various synchrotron-based measurements, and Computer and Information Network Center in National Taiwan University for providing calculation resources. This study was financially supported mainly by Taiwan's Ministry of Science and Technology (MOST) under the Academic Summit Project (106-2745-M-002-002-ASP and 107-2745-M-002-001-ASP). Financial supports from the Deep Decarbonization Project at Academia Sinica (AS-SS-106-02-3), the Center of Atomic Initiative for New Materials (AI-Mat) at National Taiwan University (107L9008 and 108L9008), and the Featured Areas Research Center Program within the framework of the Higher Education Sprout Project by the Ministry of Education (MOE) in Taiwan are also acknowledged.

## Author contributions

H-.T.L. and S-.T.C. conceived and designed the project and performed all electrochemical measurements. H-.T.L., S-.T.C., Y-.C.C., Y-.R.L., and C-.L.D. performed the in situ XAS experiments. P-.T.C. performed the theoretical calculations. H-.T.L., S-.T.C., D.P.W., C-.H.W., K-.H.C., and L-.C.C. analyzed the experimental data and prepared the manuscript. All authors reviewed and contributed to the manuscript.

## Competing interests

The authors declare no competing interests.
