## [Peer Review File · Nature Communications]

Reviewers' Comments:

Reviewer #1:

Remarks to the Author:

The manuscript studies the ORR mechanism through operando X-ray absorption and electrochemical impedance spectroscopies over a new developed Co-N-C catalyst (pyrolyzed Vitamin B12). The operando XAS in soft XAS (s-XAS) ranges, coupled with first-principle calculations, is important for providing new insight into metal active-site geometry and electronic structure evolution during ORR. However, the reviewer has some critical concerns to the results of this work.

1. The structure of the fresh catalyst used in this work is unclear. Characterizations such as SEM, TEM, HAADF-STEM, XPS, XRD, Raman of the catalysts should be provided.
2. For the operando Co L_{3,2}-edge NEXAFS results. The authors stated that "Co was moderately oxidized during catalytic reaction.", however, the author then state that "the oxidation state of Co was maintained at nearly +2 during ORR.", Why? Moreover, the conclusion should be carefully discussed in combing with the operando Co K-edge XANES results. Why there is no observed changes in the K-edge of operando XAS?
3. At 0.2 V, why does the peak of operando Co XAS spectra return to its original position? The XAS spectra of the catalyst after operando measurement should be provided.
4. The local environment of Co determined through fitting the EXAFS data is insufficiency. The Co-O species should be confirmed by more evidence. More important, why there is no changes could be observed for operando Co XAS spectra at different applied potential (1.0, 0.8, 0.6, 0.4...)?
5. Kinetic experiments should be performed to studies the rate-determining step, which is important to understand the intermediate Co-O species captured by operando XAS.
6. Figure 3 revealed the evolution of Co²⁺-oxo intermediate state. Different Co²⁺-oxo intermediates were captured at different applied potential? Why? Moreover, different Co²⁺-oxo intermediate state will also result in the changes of Co L_{3,2}-edge NEXAFS, how does the author discuss the changes of oxidation state and coordination environment of Co at the same time?
7. The English usage should be carefully polished. Many grammatical mistakes could be found even in the Abstract. For instance, "Our results revealed preferential adsorption of oxygen at the Co²⁺ center with end-on coordination forming an oxo-like species.", "the charge transfer mechanism between the catalyst and reactant enabled further realization of Co-O species formation." Too many such kind of grammatical mistakes all through the manuscript should be thoroughly revised.

Reviewer #2:

Remarks to the Author:

The manuscript presents a study of the ORR mechanism of pyrolyzed Vitamin B12 using operando X-ray absorption spectroscopy coupled with electrochemical impedance spectroscopy, which enables operando monitoring of the oxygen-binding site on the metal center. As it was claims that the results revealed preferential adsorption of oxygen at the Co²⁺ center with end-on coordination forming an oxo-like species. The topics is really interesting and would have significant impact to the catalysis research.

Overall the experimental findings are solid and convincing. However, there is consistence problem between the Co K-edge EXAFS and Co L-edge XAS conclusion. Co K-edge EXAFS reveals that the elongation of the Co-N bond was due to the distortion of the originally square planar Co-N₄/C configuration, which arose from the displacement of the Co atom from the Co-N₄/C plane induced by the oxygen-based adsorbate, such as O₂, O-O, OH, or H₂O. There is no clear evidence for the chemical bonding or adsorption of O₂, O-O, OH, or H₂O to Co metal center, as shown in figure 2(d), while Co L₃-edge XAS peak in figure 3(a) presents energy shifts being assigned to the different oxygenate species based on the FEFF calculation. There may be O₂, O-O, OH, or H₂O around Co metal center, but strength of the interaction is reflected different from these two techniques.

Also, no details on the FEFF calculation are given in regards to the pyrolyzed Vitamin B12.

It is not clear to the reviewer why the authors claimed operando X-ray absorption spectroscopy coupled with electrochemical impedance spectroscopy. There are only a few selected potentials were set to record the Co K-edge EXAFS and Co L-edge XAS.

RESPONSES SHEET

The authors are very much thankful to all the reviewers for their constructive comments on our manuscript, to make it more valuable to the scientific community. We have addressed all the issues raised by the reviewers, as detailed item-by-item below, and we believe our revised manuscript shall be satisfactory and are looking forward to the acceptance in your esteemed journal.

Response to Reviewer(s)' comments

Reviewer #1:

The manuscript studies the ORR mechanism through operando X-ray absorption and electrochemical impedance spectroscopies over a new developed Co-N-C catalyst (pyrolyzed Vitamin B12). The operando XAS in soft XAS (s-XAS) ranges, coupled with first-principle calculations, is important for providing new insight into metal active-site geometry and electronic structure evolution during ORR. However, the reviewer has some critical concerns to the results of this work.

Comment 1. The structure of the fresh catalyst used in this work is unclear. Characterizations such as SEM, TEM, HAADF-STEM, XPS, XRD, Raman of the catalysts should be provided.

Reply:

The authors are thankful to the reviewer for the important suggestion. Our group has been studying this subject for a couple of years and has reported several spectral characterizations of the py-B12 electrocatalysts, for example Raman and FTIR in Ref [1], XPS in Ref [2], and XRD in Ref [3]. Because there is no obvious change in the appearance of pyB12 and pristine B12, we did not report direct images, ie SEM or TEM. However, the operando measurements for the variation of electronic structure of the active site (CoN₄/C) during the catalytic reaction have never been reported before. Therefore, we focus on the operando spectroscopy in this study so as to better understand the origin of the catalytic process. The authors apologize for the incomplete information of the materials; we thank the reviewer's reminder, so, in the revised version we have provided a comprehensive set of ex-situ characterization results in **Figure S1**, as depicted below.

Complement in Figure S1. Ex situ characterizations of (a) Raman of py-B12 and pristine B12 [1], (b) FTIR of py-B12 and pristine B12 [1], (c) XPS of pristine B12 [2], (d) XPS of py-B12 and pristine B12 [2], (e) XRD of py-B12 and pristine B12 [3], and (f) XRD of B12 and Co₂P [3]. In Raman spectra, the two strong peaks at 1330 and 1580 cm⁻¹ are attributed to D- and G-peaks, respectively, of the carbon-like materials, suggesting that py-B12 forms a network structure of poly-aromatic hydrocarbons. Similar comparison of the FTIR spectra shows four characteristic peaks (A to D) for pristine B12. XPS suggested that the pyrolysis converts the nitrogen into pyridinic-like nitrogen and quaternary N-type

nitrogen, with N⁺ at 398.7 eV and 401.4 eV. XRD presented that pristine B12 exhibits the same characteristic peaks as observed in the vitamin B12 reference pattern.

[1] Chang, ST et al. *Energy Environ. Sci.* 2012, 5, 5305.

[2] Wang, CH et al. *RSC Advances* 2013, 3, 15375-15381.

[3] Chang, ST et al. *International Journal of Hydrogen Energy* 2012, 37, 13755-13762.

Comment 2. For the operando Co L_{3,2}-edge NEXAFS results. The authors stated that “Co was moderately oxidized during catalytic reaction.”, however, the author then state that “the oxidation state of Co was maintained at nearly +2 during ORR.”, Why? Moreover, the conclusion should be carefully discussed in combing with the operando Co K-edge XANES results. Why there is no observed changes in the K-edge of operando XAS?

Reply:

The authors greatly appreciate the reviewer for the note-worthy suggestion. The Co K-edge spectrum describes a less intense pre-edge transition from the 1s → 3d which is principally forbidden due to the selection rule. Thus, the changes of oscillation in K-edge is much less than that of L-edge (2p → 3d transition) during the ORR reaction. Compared to hard XAS (K-edge range), soft XAS (L-edge range) is a more direct and efficient tool to probe the electronic states through excitation of core-level electrons to unoccupied 3d orbitals above the Fermi level. According to the Co L_{3,2}-edge NEXAFS, the peak shifts from +2 valence number region to higher valence number region, meaning that the catalytic process is accompanied by a partial electron transfer from the 3d orbital of Co active site to adsorbed oxygen species. Therefore, we conclude that “Co was moderately oxidized during catalytic reaction” and “the oxidation state of Co was maintained at nearly +2 during ORR”.

Because Co²⁺ only moderately oxidizes from 2+ to 2+ δ (0< δ <1), the insensitive Co K-edge reveals very small change. We can still obtain the information of adsorption from the $\Delta\mu$ analysis of K-edge. The relevant discussion has been modified accordingly in **Page 10** and **Page 14** in the Result and Discussion section marked in red color.

Comment 3. At 0.2 V, why does the peak of operando Co XAS spectra return to its original position? The XAS spectra of the catalyst after operando measurement should be provided.

Reply:

The authors thank for the question. In the later stage of the reaction, oxygen species gradually turn to H₂O molecules. Due to the small interaction between H₂O and CoN₄/C site, H₂O may desorb from the catalyst surface. Therefore, the peak of operando Co NEXAFS returns to its original position, i.e. no adsorption. We have modified manuscript per the reviewer’s comment. The XAS spectra of the catalyst after operando measurement are also depicted in Figure 3(a), and more relevant discussions are added in **Page 15** in the Result and Discussion section marked in red color.

Comment 4. The local environment of Co determined through fitting the EXAFS data is insufficiency. The Co-O species should be confirmed by more evidence. More important, why there is no changes could be observed for operando Co XAS spectra at different applied potential (1.0, 0.8, 0.6, 0.4...)?

Reply:

We thank for the reviewer’s important comment. According to the fitting results, the coordination number is five at different applied potential, including four bonds to N and one bond to O. It means that single oxygen-based intermediate adsorbs on the reactive Co atom surrounded with 4 N atoms during evolution under applied potential from 1.0 to 0.4 V. The fitting data are summarized in Table 1 and Table 2 in the manuscript. Although from EXAFS, the environment around the central Co atom can be deduced, the coordination number is merely the first shell information. The information of multiple shells is still unclear. In other words, what oxygen species are bound to the Co atom is not known from the EXAFS. Therefore, we further determine the different oxygen species bonded to the central Co-N₄/C by using FEFF calculations. The Co-oxo intermediates from four-electron and two-electron processes resulted in different XANES changes. Therefore, we may conjecture what the reaction process is. Relevant sentences have been added in **Page 13** in the Result and Discussion section marked in red color.

Comment 5. Kinetic experiments should be performed to studies the rate-determining step, which is important to understand the intermediate Co-O species captured by operando XAS.

Reply:

Thanks for reviewer's constructive suggestion. We believe the in-situ XAS via quick-XANES and quick-EXAFS shall be valuable for the kinetic analysis, as suggested by the reviewer for our work. However, the relevant beamline for these time-resolved measurements in Taiwan Photon Source was newly constructed, but there is a huge challenge in implementing this state-of-art technology. We hope to keep the kinetic experiments in our next research report, yet not losing the integrity of the current manuscript.

Comment 6. Figure 3 revealed the evolution of Co²⁺-oxo intermediate state. Different Co²⁺-oxo intermediates were captured at different applied potential? Why? Moreover, different Co²⁺-oxo intermediate state will also result in the changes of Co L_{3,2}-edge NEXAFS, how does the author discuss the changes of oxidation state and coordination environment of Co at the same time?

Reply:

The authors are thankful for the crucial comment. Figure 3(a) reveals the experimental XANES. We observe that the peak shifts at different applied potential. The shift is attributed to the summation of all different Co²⁺-oxo intermediates. At different stage of applied potential, different proportions of Co²⁺-oxo intermediates are produced which result in different degree of shift. At 0.4 V and 0.6 V, the maximum redshift means a maximum amount of Co oxides.

In order to understand the origin of peak shift, we performed the theoretical calculations. In Figure 3(b), the FEFF calculations reveal that the relation between the peak shift and adsorbed oxygen species on the Co-N₄/C. The results suggest that the adsorption of O atom on Co leads to largest redshift. It implies that there are largest amount of O atom adsorption at 0.4 V and 0.6 V. The O atom can be evolved from direct dissociation of O₂ through the four-electron process. Comparing to the calculated results of two-electron process provided in supporting information S4 (O₂, H₂O₂, OH, and H₄O₂), we find that none of these species leads to such a large oxidation state. These results suggest that the reaction obeys the four-electron process. We have revised the relevant sentences in **Page 14** and **Page 15** in Result and Discussion section, marked in red color.

Comment 7. The English usage should be carefully polished. Many grammatical mistakes could be found even in the Abstract. For instance, "Our results revealed preferential adsorption of oxygen at the Co²⁺ center with end-on coordination forming an oxo-like species.", "the charge transfer mechanism between the catalyst and reactant enabled further realization of Co-O species formation." Too many such kind of grammatical mistakes all through the manuscript should be thoroughly revised.

Reply:

The authors deeply apologize for the mistakes. The whole manuscript has been thoroughly checked for English grammar mistakes.

Reviewer #2:

The manuscript presents a study of the ORR mechanism of pyrolyzed Vitamin B12 using operando X-ray absorption spectroscopy coupled with electrochemical impedance spectroscopy, which enables operando monitoring of the oxygen-binding site on the metal center. As it was claimed that the results revealed preferential adsorption of oxygen at the Co²⁺ center with end-on coordination forming an oxo-like species. The topic is really interesting and would have significant impact to the catalysis research.

Overall the experimental findings are solid and convincing. However, there is a consistency problem between the Co K-edge EXAFS and Co L-edge XAS conclusion. Co K-edge EXAFS reveals that the elongation of the Co–N bond was due to the distortion of the originally square planar Co–N₄/C configuration, which arose from the displacement of the Co atom from the Co–N₄/C plane induced by the oxygen-based adsorbate, such as O₂, O–O, OH, or H₂O. There is no clear evidence for the chemical bonding or adsorption of O₂, O–O, OH, or H₂O to Co metal center, as shown in figure 2(d), while Co L₃-edge XAS peak in figure 3(a) presents energy shifts being assigned to the different oxygenate species based on the FEFF calculation. There may be O₂, O–O, OH, or H₂O around Co metal center, but strength of the interaction is reflected differently from these two techniques.

Also, no details on the FEFF calculation are given in regards to the pyrolyzed Vitamin B12.

It is not clear to the reviewer why the authors claimed operando X-ray absorption spectroscopy coupled with electrochemical impedance spectroscopy. There are only a few selected potentials were set to record the Co K-edge EXAFS and Co L-edge XAS.

Reply:

The authors are gratified by the suggestion of the reviewer. We have made significant modifications regarding the reviewer's concern. Hopefully, the revisions have shown satisfactory improvements in the overall quality of the manuscript.

According to the EXAFS fitting of Co K-edge shown in Figure 2(c), the prominent peak at $\sim 2 \text{ \AA}$ is attributed to the adsorption of oxygen species at Co–N₄/C. The adsorption results in the increase of Co–N bond. The fitting analysis also provides the information of the coordination number of oxygen species on the Co atom. Figure 2(d) shows the fitting results, suggesting one oxygen species bonded to Co. EXAFS allows us to obtain the first-shell environment around the reactive Co atom. However, we could not obtain the information of multiple shells. In other words, we have no idea about what kind of oxygen species adsorbed on Co. Therefore, we further investigated the possible oxygen species bonded to the Co–N₄/C by using FEFF calculation.

From Figure 3(a), we observe that XANES has obvious shift during reaction, which represents the change of Co oxidation number. Co²⁺ moderately oxidizes from 2+ to 2+ δ ($0 < \delta < 1$). In order to understand what adsorption may cause this peak shift, we performed the theoretical calculation. In Figure 3(b), the FEFF calculations reveal that the relation between the peak shift and oxygen species adsorbed on the Co–N₄/C. The results suggest that the adsorption of O atom on Co leads to largest redshift. However, the shift in experimental spectrum is attributed to the summation of all different Co-oxo intermediates. Here, we can suggest that there are largest amount of O atoms adsorption at 0.4 V and 0.6 V. The O atom is generated from direct dissociation of O₂ through the four-electron process. In contrast, there is no Co-oxo intermediates in two-electron process that can significantly increase the Co oxidation state. The calculation results of the both processes have been provided in Figure S4 (O₂, H₂O₂, OH, and H₄O₂) and below. These results suggest that the reaction obeys the four-electron process.

XANES calculations were performed using DFT-optimized structures to theoretically examine Co L-edge XANES spectra related to oxygen-based adsorbates on the Co–corrin cluster. The optimized structures are shown below (**Figure S5**). XANES calculations of Co were performed by the FEFF8 code. The self-consistent potential and full multiple scattering were calculated at a 5.0- \AA radius. In the initial input, a negative charge is equally allocated on every atom of a cluster. After self-consistent calculation, the charge would be redistributed correctly. To compare the experimental and calculated spectra, a rigid shift of 2 eV to higher energy was applied to each calculated spectrum. These FEFF calculation details have been added in the supporting information **Figure S5**.

Complement in Figure S5. (a) Optimized structures for the models of four-electron process. (b) FEFF calculated Co L-edge with the different oxygen species of four-electron process. (c) Optimized structures for the models of two-electron process. (d) FEFF calculated Co L-edge with the different oxygen species of two-electron process. If the reaction went through two-electron process, there would be no redshift in the XANES spectrum.

We have combined XAS and EIS measurement under in situ conditions in our current manuscript. The EIS spectrum provides the information of reaction stages and characterization of bulk and interfacial properties which give us a better understanding of the macroscopic view of our catalyst. So that, three stages, the mix diffusion controlled region (in approximately 1.1–0.9 V), kinetic-dominant region (in 0.8–0.4 V), and mass transport region (in 0.3–0 V) are clearly differentiated. Furthermore, the XAS measurements offer better understanding of microscopic view. The EXAFS highlights the local structure. The XANES infers the variation of oxidation state. All the changes are associated with the evolution of reaction stages. Therefore, the insight of atomic level into the reaction can be gained. Also, by combining our FEFF calculation, we can simulate the molecular orbital under the reaction pathway. We can get the benefit from those measurements.

Reviewers' Comments:

Reviewer #1:

Remarks to the Author:

The revised manuscript has adequately addressed all key concerns raised by the reviewers. I therefore recommend for the publication in Nature Communications.

Reviewer #2:

Remarks to the Author:

The responds to the review comments are proper and informative at a satisfying level. Reviewer likes to see more clear discussion on the comparison between Co L_{2,3}-edge XAS and Co K-edge XANES in regards to the intrinsic characterization specs of soft x-rays and hard x-rays, surface sensitivity (5-10 nm) vs bulk sensitivity (> microns), and the chemical compositions.

RESPONSES SHEET

Reviewer #1 (Remarks to the Author):

The revised manuscript has adequately addressed all key concerns raised by the reviewers. I therefore recommend for the publication in Nature Communications.

Reply:

The authors greatly appreciate the reviewer for the note-worthy suggestions for improving the whole manuscript.

Reviewer #2 (Remarks to the Author):

The responds to the review comments are proper and informative at a satisfying level. Reviewer likes to see more clear discussion on the comparison between Co L2,3-edge XAS and Co K-edge XANES in regards to the intrinsic characterization specs of soft x-rays and hard x-rays, surface sensitivity (5-10 nm) vs bulk sensitivity (> microns), and the chemical compositions.

Reply:

The authors are gratified by the suggestion of the reviewer. The figure below displays the correlation of the length of X-ray penetration and photon energy, where the curve going down means X-ray being absorbed. For instance, pristine B12, the Co L-edge is obtained at a depth of 1 μm , as well as the K-edge is obtained at ca. 100 μm . Therefore, the spectral signal is not only the surface adsorption state, but also part of the bulk information. K-edge contains a large proportion of bulk information, which is not conducive to the analysis of adsorbates. Therefore, the $\Delta\mu$ analysis of K-edge is used to discuss the adsorption of py-B12. The discussion has been added at the end of the first paragraph of “Operando X-ray absorption and electrochemical impedance measurements” in the Method section.

Figure. X-ray attenuation length. (https://henke.lbl.gov/optical_constants/atten2.html; available online)